# Adipose Tissue Derivatives in Peripheral Nerve Regeneration after Transection: A Systematic Review

**DOI:** 10.3390/bioengineering11070697

**Published:** 2024-07-10

**Authors:** Rafael Silva de Araújo, Matheus Galvão Valadares Bertolini Mussalem, Gabriel Sant’Ana Carrijo, João Victor de Figueiredo Bani, Lydia Masako Ferreira

**Affiliations:** 1Federal University of São Paulo, Department of Plastic Surgery, São Paulo 04038-001, Brazil; matheus.bertolini@unifesp.br (M.G.V.B.M.); jvfbani@unifesp.br (J.V.d.F.B.); lydiamferreira@gmail.com (L.M.F.); 2FMABC University Center, Faculty of Medicine, Santo André 09060-870, Brazil; gabriel.carrijo@aluno.fmabc.net

**Keywords:** adipose tissue derivatives, adipose stem cells, microsurgery, nerve repair, nerve regeneration, functional analysis

## Abstract

Introduction: Peripheral nerve injury (PNI) is increasingly prevalent and challenging to treat despite advances in microsurgical techniques. In this context, adipose tissue derivatives, such as adipose-derived stem cells, nanofat, and stromal vascular fraction have been gaining attention as potential allies in peripheral nerve regeneration. Objectives: This study aims to explore the use of adipose tissue derivatives in nerve regeneration following peripheral nerve transection in murine models. Thus, we assess and synthesize the key techniques and methods used for evaluating the obtained nerve regeneration to guide future experimental research and clinical interventions. Methodology: A systematic review was conducted in February 2024, adhering to the Cochrane and PRISMA 2020 guidelines, using the PubMed, SciELO, and LILACS databases. The focus was on experimental studies involving adipose tissue derivatives in nerve regeneration in animal models post-transection. Only experimental trials reporting nerve regeneration outcomes were included; studies lacking a comparator group or evaluation methods were excluded. Results: Out of 273 studies initially identified from MEDLINE, 19 were selected for detailed analysis. The average study included 32.5 subjects, with about 10.2 subjects per intervention subgroup. The predominant model was the sciatic nerve injury with a 10 mm gap. The most common intervention involved unprocessed adipose-derived stem cells, utilized in 14 articles. Conclusions: This review underscores the significant potential of current methodologies in peripheral nerve regeneration, particularly highlighting the use of murine models and thorough evaluation techniques.

## 1. Introduction

Peripheral nerve injury (PNI) has been garnering increased attention due to its rising prevalence alongside rapid societal developments [1]. The main causes of PNI are related to damage to nervous structures, such as traumatic injuries, which can present in diverse clinical manifestations. The reviews of the literature indicate a prevalence of 8% for non-traumatic cases [2] and 5% for traumatic instances [3]. Despite advancements in microsurgical techniques, patients with PNI still face unsatisfactory recovery prognosis [4].

Injuries are categorized into neuropraxia, axonotmesis, and neurotmesis, each with distinct implications for treatment and functional recovery [5,6]. This scenario underscores the need for innovative therapeutic approaches that could offer more promising outcomes.

In this context, adipose stem cells (ASCs) emerge as a promising source for regenerative therapies due to their ability to differentiate into various cell types, such as Schwann-like cells, and to release regenerative factors through paracrine mechanisms [7,8].

The scientific literature highlights the potential of these cells in treating peripheral nerve injuries in animal models, with evidence of their role in promoting regeneration and improving nerve function. Experimental studies have been primarily conducted in rodent models, as their nerves share similar size, fascicular organization, and morphology to humans [9]. Regarding the use of ASCs in treatment, the potential to contribute to the recovery of damaged nerves makes them promising candidates for future clinical applications [10].

Additionally, complementary techniques such as the stromal vascular fraction (SVF) and nanofat are explored to optimize the regenerative potential of ASCs with paracrine and immunomodulatory actions to induce tissue regeneration. These approaches combine the cellular richness of the SVF with the benefits of the extracellular matrix and paracrine signals provided by nanofat, enabling the development of more effective treatments against nerve injuries [11,12].

In this context, this systematic review aims to present the state of the art in the current literature regarding the use of murine models subjected to transection injuries in the peripheral nerves and that have utilized adipose tissue derivatives to promote nerve regeneration. Subsequently, it intends to synthesize the main methods of analyzing nerve regeneration and their respective results in the selected articles.

## 2. Materials and Methods

### 2.1. General Information

This is a systematic search study following the Cochrane standards, and this manuscript was completed according to the Preferred Reporting Items for Systematic Reviews and Meta-Analyses (PRISMA) 2020 guidelines [13], with data research conducted in February 2024. As this is a systematic literature review, this study was exempt from institutional review board approval and did not utilize external funding sources. The databases PubMed, Scielo, and LILACS were used to search for articles addressing peripheral nerve regeneration in animals subjected to nerve transection. The following keywords were used in PubMed: (“rat” [All Fields] OR “mice” [All Fields] OR “rabbits” [All Fields] OR “macaca” [All Fields] OR “swine” [All Fields] OR “muridae” [All Fields]) AND (“peripheral nerve injuries” [All Fields] OR “denervation” [All Fields]) AND (“microsurgery” [All Fields] OR “neurosurgery” [All Fields] OR “mesenchymal stem cells” [All Fields] OR “adipose tissue” [All Fields]) AND “nerve regeneration” [All Fields]. The same terms were used in SciELO and LILACS with necessary modifications according to the rules of those databases.

The search for articles on the topic was conducted to encompass the largest number of studies, aiming to portray the current state of the art regarding this type of experimental assay. This allows for the evaluation of the variation in academic production on this subject in a historical panorama. Articles from January 1969 to December 2023 were selected.

### 2.2. Data Selection and Extraction

Following data research, article selection was conducted by three independent reviewers in two stages: analysis of article titles and abstracts, followed by full-text review. Any conflicts were resolved by a third author. The literature was restricted to articles in English, Portuguese, and Spanish without date limitations. Inclusion criteria were limited to articles assessing peripheral nerve regeneration as the primary outcome and considering the section of the peripheral nerve studied. Moreover, only experimental trials were included since they are the primary studies with the highest impact, thus providing more reliability for the data the authors aimed to analyze. The exclusion criteria were defined as follows: articles that did not address the presence of a comparator group among the studied subgroups; those that did not include methods for the evaluation of the studied nerve regeneration; and those that did not use adipose derivatives in at least one subgroup. In the present study, adipose tissue derivatives were considered, including fat graft, ASCs, and raw materials obtained mechanically (nanofat and microfat) and chemically (stromal vascular fraction). After applying the inclusion and exclusion criteria, the studies were included for full-text analysis. It is important to note that the choice of inclusion and exclusion criteria was made a priori.

Regarding data extraction, it was performed by three independent reviewers who collected the following information from the articles: author, publication year, intervention, objective, conclusion, study location, animal follow-up, population, number of animals per subgroup, breed of animal, functional analyses, histological analyses, and other parameters assessing nerve regeneration. Data on funding sources were also collected.

## 3. Results

The literature search yielded 273 records, mainly from the MEDLINE database. Abstract screening removed the majority of the articles. The main causes for removal were wrong study design and studies that did not use adipose tissue byproducts. More details can be found in Figure 1.

The country in which the institution of the first author is located was considered as the country of origin of the article. The country that produced the most results was China (*n* = 4), followed by Switzerland (*n* = 3). A complete breakdown of the countries can be found in Figure 2.

Despite including several animal species in our search terms, we only located studies conducted in muridae. The most commonly used species was the Sprague Dawley rat, followed by Wistar and Lewis rats. FVB rats and mice were used in a single study each. The average number of subjects was 32.5 (SD = 17.9), being the average number of subjects per intervention group 9.2 (SD = 5.2)

Most articles performed their intervention immediately after neurotmesis, and then observed the rat for a mean number of 11.26 (SD = 5.4) weeks.

The sciatic nerve injury with a gap of 10 mm was the overall most commonly utilized model by a distinct margin (*n* = 11). Other gaps were utilized in smaller nerves or animals. One study utilized a chronic denervation model by severing the peroneal nerve and suturing the ends away from each other. Figure 3 presents a breakdown of all models.

A total of 38.8%. of control groups utilized empty conduits. All articles utilized a control group in accordance with the intervention.

ASCs with no further processing were the most common intervention, utilized in a total of 14 articles and 122 nerves. A total of 68.4% of the articles utilized conduits to bridge the nerve gaps. Aside from ASCs, other adipose tissue byproducts utilized were SVF and fat grafts. A breakdown of all results can be seen in Table 1.

## 4. Discussion

Based on the current research, tissue bioengineering and microsurgery emerge as some of the fundamental pillars of regenerative medicine. As intervention studies on vital structures progress, the techniques described increasingly approach practical applicability.

### 4.1. General Aspects

Among the selected studies, China contributed the highest number (*n* = 4), followed by Switzerland (*n* = 3). This find is compatible with the recent advances that China has been making in the academic publishing scene [33]. Remarkably, Chinese articles had the second-highest average journal impact factor, surpassed only by Korea. This indicates that China has had an increase not only in the quantity, but also in the quality of their publications.

### 4.2. Adipose Tissue Derivatives

In accordance with our inclusion criteria, all articles involved ASC-based interventions in at least one group, totaling 269 nerves undergoing sectioning and ASC interventions. The use of ASCs has been gaining popularity within the literature since its discovery as a potential material for tissue regeneration [34]. A search on PubMed for ‘Adipose-derived stem cells’ shows an increasing trend, indicating the growing interest in this topic. Several uses of ASCs are currently under investigation, with the main ones including wound healing, bone healing, immunomodulation, and nerve regeneration [35]. ASCs have gained special attention in the field of stem cell research because they are widely accessible throughout the body, easy to process, and lack the ethical controversies associated with other stem cell sources [36].

The first article proposing the use of ASCs for neuronal regeneration was published in 2005, followed by numerous animal and human research models exploring its therapeutic potential [37]. Their applications across various specialties, including ophthalmology (optic nerve regeneration), urology (treatment of nerve-induced erectile dysfunction), and reconstructive plastic surgery, have been extensively studied [38,39,40].

### 4.3. Trauma Mechanism

The studies on ASCs in nerve regeneration remain predominantly confined to animal models, notably involving the sciatic nerve neurotmesis with a 10 mm gap. Among the analyzed articles, 11 studies employed this specific model, while others explored neurotmesis involving different gaps or other nerves, such as the brachial plexus and the facial nerve. Additionally, several studies investigated crush injury models [41], although they were not included in this review.

The selected studies exhibited a broad spectrum of methodologies and approaches, leading to considerable variability in the types of functional and histological analyses performed. This diversity hindered the feasibility of conducting a meta-analysis, which generally demands a more standardized approach to data synthesis and analysis [42]. Consequently, we adopted a systematic review methodology to provide a comprehensive and interpretive synthesis of the existing literature.

### 4.4. Functional Analysis

In this review, eight articles [16,19,20,21,22,25,26,31] utilized the “Sciatic Functional Index” (SFI) for evaluating nerve function. Based on Medicaneli et al. (1982) [43], the method is reliable for assessing the muscle strength index and tissue electrophysiology, thus evaluating the performance of nerve function in the injured lower limb. For instance, the regeneration of the sciatic nerve was assessed by Dai et al. (2013) [19] with recovery rates ranging from 20.4% to 23.4% over a three-month period. Similarly, Orbay et al. (2012) [44] reported a functional recovery rate of 24.9% within two months in mice that received the intervention, demonstrating the method’s effectiveness in analyzing this variable [45,46,47,48]. Conversely, other studies also used alternative scales. Mohammadi et al. (2015) [22], in addition to the SFI, employed the “Basso, Beattie, and Bresnahan for limb motor function” (BBB) and “Static Sciatic Index” (SSI) for assessing the same parameter. Although less used in the reviewed articles, both have validation in the scientific literature for their use [49,50]. Only one article used the “Peroneal Functional Index” (PFI), although the literature equally validates its use compared to the SFI [51].

Regarding methods of nerve electrostimulation, the parameter compound muscle action potential (CMAP) was used in five articles [15,17,18,28,30], compound nerve action potential (CNAP) in one article [25], and nerve conduction velocity (NCV) in four articles [16,19,20,25].

The CMAP is intended to measure the sum of the action potential of the concerned muscle fibers, focusing on the response of the innervated muscle [52]. Notably, the experiment by Di Summa et al. (2011) [15] assessed nerve regeneration through significant increases in axonal diameter and CMAP using either autograft or fibrin seeded with ASCs on the PNI. The validation of this method is also present in the literature [53].

The use of CNAP also proved to be a satisfactory parameter in the study of He et al. (2016) [25], being effective in comparing the proposed regeneration between two groups that used ASCs obtained from different sources. This suggests its potential in differentiating the quality of nerve regeneration even with interventions of a similar nature.

The measurement of nerve conduction velocity is also an important factor for evaluating innervation of the site, and depends on factors such as axon diameter, myelin sheath thickness, and the length of the “internodes” [54,55]. According to Raisi et al. (2014) [20], NCV was effective in ascertaining poorer performance in nerve regeneration in the pro-inflammatory MSC group, with a lower conduction velocity rate (72%). This aspect was crucial in questioning which factors were triggering, such as smaller diameter of regenerating axons, thinner myelin sheaths, or even an immaturity of myelinated nerve fibers in that group.

Regarding the muscle weight ratio (MWR), nine articles used this method [17,19,21,22,23,25,26,29,31]. This parameter appears to be used due to the ease of its application in studies, often being the only type of functional evaluation performed [23,29]. In the experimental trial conducted by Wang et al. (2023) [41], the method proved effective in comparing nerve regeneration in the transection versus crushing of the peripheral nerve group.

### 4.5. Histological Analysis

The use of optical microscopy alongside software for image analysis was highlighted in all selected articles. Based on Kappos et al. (2015) [21], this combined method was crucial for measuring axonal diameter and fiber diameter. Thus, the thickness of myelin and other important average parameters for evaluating local tissue regeneration can be calculated and compared across groups [56].

The light microscope method was enhanced by complementary analyses in several studies. Notably, the use of toluidine blue in five articles [22,23,25,27,29] emphasized its role in staining myelinated structures that facilitate neuronal visualization [57,58,59]. According to Ozkan et al. (2016) [27], this method proved effective for verifying regular myelinated pre-neuronal bodies with an increased diameter in the study group, demonstrating better organization of nerve tissue regeneration.

Previous articles have successfully utilized immunofluorescence methods for analyzing the regeneration of Schwann cells and peripheral remyelination [60,61]. In agreement, Kastamoni et al. (2023) [32] found that the expression of EphA4 was associated with a negative effect on the axonal repair of the sciatic nerve, consistent with previous studies [62,63,64].

Despite its requirement for considerable time and specialized training, transmission electron microscopy is widely mentioned in the literature as a method for obtaining detailed, high-quality images of nervous structures [65,66,67,68]. In this regard, several studies have used this method to obtain highly specific images of structures [16,18,20,25,28,29,30].

### 4.6. Other Analysis

Tremp et al. (2015) [24] identified a correlation between axonal length measured by MRI and axonal length measured by immunohistochemistry, reaffirming the nerve damage between the sciatic nerve of the intervention versus control. Thus, it is understood that the use of this method can add comparative value between the analyses performed.

Flow cytometry is an effective tool in the studies of axonal nerve regeneration, due to its ability to quantitatively analyze individual cells in large populations [69,70,71]. One study used this method to evaluate specific cellular groups and their tissue behavior in regeneration [29].

RNL is a prospective method for evaluating nerve regeneration that compares neuronal expansion through its connections [72,73,74]. Di Summa et al. (2011) [15], using this method and electrostimulation, evaluated that the autograft group and the ASCs group showed greater motoneuron regeneration compared to the control, which is in concurrence with the current literature [75,76,77].

### 4.7. Strengths and Limitations

This review fulfills its role of assessing the current state of the scientific literature regarding the use of murine models for the experimentation of peripheral nerve regeneration, as well as elucidating and analyzing the main functional assessment models used in such experimental trials. Another point analyzed was the global distribution of scientific production for these article models, with an emphasis on comparing the use of different adipose tissue derivatives, such as ADSCs, SVF, and nanofat. However, although there are well-defined study populations and outcomes, the breadth of types of peripheral nerve injury makes the execution of a meta-analysis unfeasible, due to the heterogeneity of the selected articles concerning the type of intervention proposed in each study. Lastly, the heterogeneity of the subgroups and evaluation methods complicates a more sophisticated biostatistical analysis.

## 5. Conclusions

The techniques and procedures described in this review hold significant potential for future practical application in peripheral nerve regeneration, as further studies continue to advance the field. Thus, this work serves its purpose by elucidating the current evidence on the topic, evaluating the principal models and formats of experimental studies that permeate the literature. Additionally, it enhances the clarity of the main types of adipose tissue derivatives and the methods for evaluating nerve regeneration in the most methodologically rigorous articles reviewed.

## Figures and Tables

**Figure 1 bioengineering-11-00697-f001:**
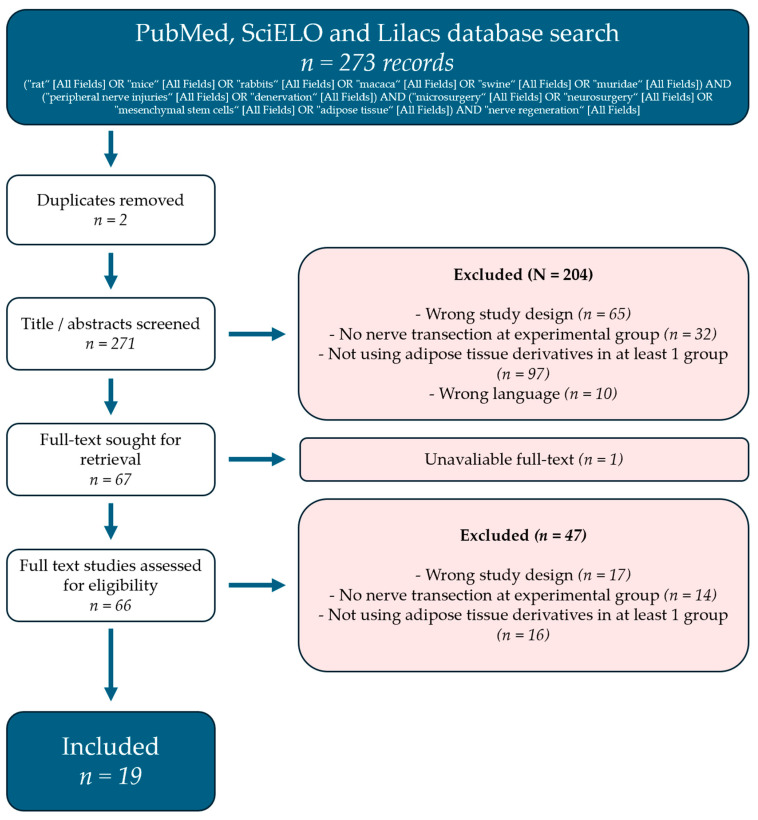
PRISMA flow diagram representing the review selection process.

**Figure 2 bioengineering-11-00697-f002:**
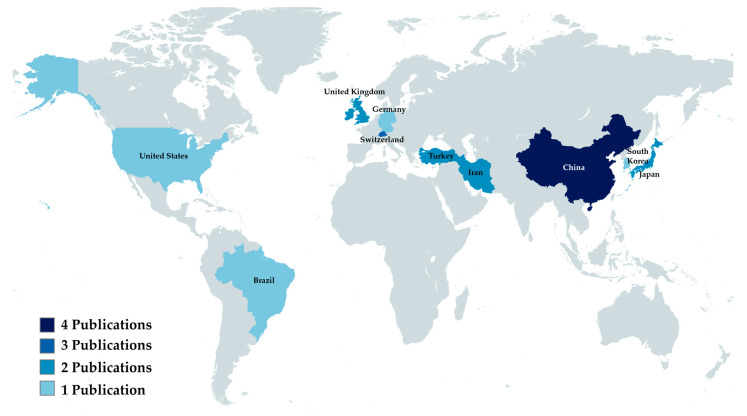
Analytical view of publishing countries.

**Figure 3 bioengineering-11-00697-f003:**
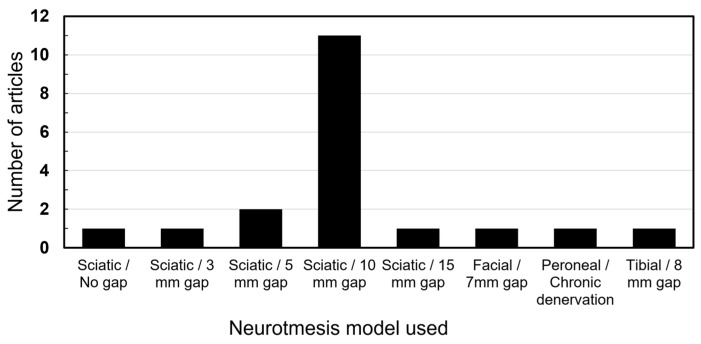
Nerve injury models utilized.

**Table 1 bioengineering-11-00697-t001:** Table containing breakdown of the characteristics of each study.

Author and Year	Sample	Study Groups	Functional Analysis	Histological Analysis	Other Analyses
Reid et al. (2011) [14]	12	Artificial CD; CD + ASCs; CD + SCs	-	LM; IAS; and Immunofluorescence	PCR; Western Blot; and ELISA
di Summa et al. (2011) [15]	25	Fibrin CD; Fibrin CD + ASCs; Fibrin CD + SC; Fibrin CD + BMSC; Autograft	CMAP	LM; IAS; and Immunofluorescence	RNL
Liu et al. (2011) [16]	30	Acellular nerve graft + DMEM; Acellular nerve graft + ASCs; Autograft	SFI and NCV	LM; IAS; TEM; and Immunofluorescence	PCR
Tomita et al. (2012) [17]	18	Vehicle; ASCs; SCs	PFI; CMAP; and MWR	LM; IAS; and Immunofluorescence	-
Gu et al. (2012) [18]	30	Silicone CD + fibrin glue; CD + Undifferentiated ASCs; CD + neurally differentiated ASCs	CMAP	LM; IAS; TEM; and Immunofluorescence	Western Blot
Dai et al. (2013) [19]	30	CD; CD + ASCs; CD + SCs; CD + DPSC; CD + ASCs + SCs; CD + DPSC + SCs	SFI; NCV; and MWR	LM and IAS	PCR
Raisi et al. (2014) [20]	90	CD; ASCs undifferentiated; ASCs microvesicles; ASCs Microvesicles + Pro-inflammatory; ASCs Microvesicles + anti-inflammatory	SFI and NCV	LM; IAS; and TEM	-
Kappos et al. (2015) [21]	49	CD + ASCs undifferentiated; CD + SCs; CD + autograft; CD + Human SVF; CD + Differentiated ASCs; CD + Human superficial ASCs; CD + Human Deep ASCs	SFI and MWR	LM and IAS	MRI scanning
Mohammadi et al. (2015) [22]	30	Sham; autograft; Allograft + SVF	BBB; SFI; SSI and MWR	LM; IAS; TBS; and Immunofluorescence	Biomechanical Stretch Testing
Reichenberger et al. (2015) [23]	50	Fibrin glue; Fibrin glue + ASCs	MWR	LM; IAS; TBS; and Immunofluorescence	-
Tremp et al. (2015) [24]	49	CD + DMEM; CD + ASCs; differentiated ASCs (SC-like); Human superficial ASCs; Human Deep ASCs; Human SVF; SCs	-	LM; IAS; and Immunofluorescence	MRI scanning
He et al. (2016) [25]	16	ASCs; ASCs with transduced vector control	SFI; CNAP; NCV; and MWR	LM; IAS; TBS; TEM; and Immunofluorescence	PCR
Sowa et al. (2016) [26]	20	CD; ASCs; SCs; Sham	SFI and MWR	LM; IAS; and Immunofluorescence	-
Özkan et al. (2016) [27]	30	Vein graft; Vein graft + SVF; Autograft	-	LM; IAS; and TBS	-
Shimizu et al. (2018) [28]	24	CD; CD + ASCs; CD + SVF	CMAP	LM; IAS; and TEM	RNL
Chen et al. (2019) [29]	28	Vehicle; ASCs	MWR	LM; TEM; IAS; TBS and Immunofluorescence	FC; PCR and Western Blot
Durço et al. (2020) [30]	30	CD + DMEM; CD + ASCs	CMAP	LM; TEM and IAS	Pinprick Test and RNL
Schilling et al. (2023) [31]	Unknown *	Control; Nanofat	SFI; MWR and TCF	LM and IAS	PCR; Cytokine Quantification
Kastamoni et al. (2023) [32]	24	Control: neurotomy of the left sciatic nerve; Intervention: fat graft at right nerve	-	LM; IAS; and Immunofluorescence	-

Observations: All studies divided the sample equally between groups. * Article did not mention sample size. Abbreviations: CD, conduit; ASCs, adipose stem cells; SCs, Schwann cells; BMSC, bone marrow-derived mesenchymal stem cell; DMEM, Dulbecco’s Modified Eagle Medium; SVF, stromal vascular fraction IAS, image analysis system; LM, light microscope; SFI, Sciatic Functional Index; PFI, Peroneal Functional Index; MWR, Muscle Weight Ratio; NCV, nerve conduction velocity; CMAP, compound muscle action potential; CNAP, Compound nerve action potential; TEM, transmission electron microscopy; BBB, “Basso, Beattie, and Bresnahan” for limb motor function; SSI, Static Sciatic Index; TCF, tetanic contraction force; TBS, toluidine blue staining; RNL, retrograde neuronal labelling; FC, flow cytometry; PCR, polymerase chain reaction.

## Data Availability

The raw data supporting the conclusions of this article will be made available by the authors on request.

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
