# Peer review of "Adipose Tissue Derivatives in Peripheral Nerve Regeneration after Transection: A Systematic Review"

_bioengineering, 2024, doi:10.3390/bioengineering11070697_

Round 1
Reviewer 1 Report
Comments and Suggestions for Authors
This is a systematic review aimed to explore the use of murine models in peripheral nerve transection research, evaluating and synthesizing key methods for analyzing nerve regeneration to guide future research and clinical interventions. Main results based on 19 selected studies were that sciatic nerve injury with a 10 mm gap was the predominant model and the most common intervention was unprocessed adipose-derived stem cells. The authors concluded that the review underscores the significant potential of current methodologies in peripheral nerve regeneration. Please refer to the following:
Abstract: 10.16 change with 10.2
Introduction: for clarification, injury is usually subjected to acute trauma, e.g. nerve transection. Diabetes mellitus, autoimmune diseases and infections are systemic diseases, usually affected multiple nerves (i.e. neuropathy or polyneuropathy). Since the theme of this review is regeneration after the nerve transection, it would be most appropriately to stick with that.
Materials and methods: First, SciELO and Embase, please change. Second, databases in general information do not match with figure 1.
Figure 1: please change N with n. Box with experimental studies… - n space = space 19.
Figure 2 and 3: in my opinion these two figures do not add to understanding of the manuscript. Please consider to delete figures or move to supplemental material. The same question arises with figure 4, 5 and 6.
Discussion: at the beginning of the discussion, please bring in front the main findings.
Author Response
Dear Reviewer 1,
Thank you for your thorough review and valuable feedback on our manuscript titled "Adipose Tissue Derivatives in Peripheral Nerve Regeneration After Transection: A Systematic Review." We have carefully considered each of your suggestions and have made the necessary revisions to address the issues raised. Below, we provide detailed responses to your comments and outline the specific changes made to the manuscript.
- Abstract: 10.16 change with 10.2
- Modification made as requested.
- Introduction: for clarification, injury is usually subjected to acute trauma, e.g. nerve transection. Diabetes mellitus, autoimmune diseases and infections are systemic diseases, usually affected multiple nerves (i.e. neuropathy or pol-yneuropathy). Since the theme of this review is regeneration after the nerve transection, it would be most appropriately to stick with that.
- We modified the introduction passage that confused neuropathy with nerve injury, making it clear that the study focuses on structural nerve lesions.
- Materials and methods: First, SciELO and Embase, please change. Second, databases in general information do not match with figure 1.
- We adjusted the databases in Figure 1; indeed, there was an inconsistency error with the description of the databases utilized.
- Figure 1: please change N with n. Box with experimental studies… - n space = space 19.
The figure 1 was adjusted with the suggested formatting and exclusion reasons were listed for better comprehension.
- Figure 2 and 3: in my opinion these two figures do not add to understanding of the manuscript. Please consider to delete figures or move to supplemental material. The same question arises with figure 4, 5 and 6.
We have reconsidered the purpose of each Figure and have eliminated Figures 2, 4, 5, 7, 8, and 9. We hope that the addition of the new “Table 1” will be a more interesting approach. Figure 3 was optimized for readability.
- Discussion: at the beginning of the discussion, please bring in front the main findings.
- Modifications have been made comprehensively throughout the paper to better elucidate the main objectives and findings of the study. In the results section, the findings were more specified, and a new "Table 1" was added, which lists the studies and their main findings regarding nerve regeneration assessment models. The discussion is clearer, with its main findings presented in topics of interest, with the beginning encompassing the current state of the art in nerve regeneration. Following this, the main findings of the study are presented. We hope that the changes made have made these points clearer.

Reviewer 2 Report
Comments and Suggestions for Authors
This manuscript is a review of repair strategies for peripheral nerve regeneration after nerve transection. The authors state that they used the appropriate Cochrane guidelines for such a review. They then searched three data bases for articles. They found 273 articles and then selected only 19 articles. They state that: “The most common intervention involved unprocessed adipose-derived stem cells, utilized in 14 articles.
Concerns:
1. The title and the abstract are very misleading and do not represent what was done. Both suggest that the authors are looking at ALL types of repair strategies for peripheral nerve repair. And then their wording suggests that they found that studies using adipose-derived stem cells (I will abbreviate ASCs) were “most common”. This implies that they looked for papers that used all types of nerve repair strategies and then within this, they found that the most used ASCs. Instead, they purposefully set out to search ONLY for ASCs in their search terms. This needs to be made clear in both the title and the abstract.
2. What was the purpose of this review? The authors do not describe the criteria that they used to go from 273 articles down to 19. This has to be given, to understand why this review was written.
3. Did the authors only look at ASCs? In Fig. 8 they list many different types of interventions. This includes one article that used BMSCs. There is a very extensive literature on BMSCs and their use in nerve repair. The same for MSCs in general and ASCs are one type of MSCs. If these fields were indeed included in their search, there is no way that they could come up with just one article. The search strategy did indeed include MSC as a term. But that is not one intervention listed in the graphs. ASCs are one type of MSC, and are often referred to as MSCs.
4. Over what time span did the authors search? The authors say indefinite time frame, but they should give a more definitive time frame. Presumably they started in 2005. When did the search go until?
5. This is definitely NOT a thorough identification of all the articles that have used ASCs for peripheral nerve repair. Just a few examples are:
a. A recent review of adipose tissue stem cells in peripheral nerve regeneration (Rhodes et al., J Neurosci Res. 2021;99:545–560. https://doi.org/10.1002/jnr.24738) lists Rhodes, cites 17 papers between 2016 and 2021. So if they could find 17 over 5 years, how did the authors find only 19 over a longer time period? Presumably the authors started their search in 2005, which is when they say that the first lab did this (Kokai et al. in 2005 from the K. Marra lab). (this is not my lab).
b. The authors cite papers from well-known and prestigious labs: from K. Marra’s lab (Ref 37, Kokai et al.) and from the lab that included Di Summa, Kalbermatten and Terenghi, together at one point in Switzerland. Each of these three authors then separated, and some moved to other countries. However, only one paper is cited from, for example, Di Summa. All of these authors have multiple publications that describe ASCs in nerve repair. Example: an excellent review of the potential of ASCs for use in nerve regeneration was published in 2013 with Terenghi as one author. (These are not my labs).
6. The authors state in the Discussion, page 7, line 94, that a search on PubMed for ASCs shows an increasing trend of articles. Why didn’t the authors use PubMed as one of their sources of articles?
7. Section 4.2 discusses Fat Grafting and Stem Cells. There is an extensive literature that explores the use of grafting pieces of fat over and around nerve conduits for improving nerve repair. Is this what the authors are talking about?
8. Figures 8 and 9 are almost identical. But I do not understand what the difference is between them. Especially Fig. 9. It supposedly lists: Materials used in nerve repair by “total number of nerves severed.” What does that mean?
9. The graphs are confusing in other ways. There are two colors for articles that presumably used conduits and others that did not. In Fig. 7, one paper used autografts as the control condition, but the autograft bar is labeled orange, meaning, I suppose, that it used conduits? Do you mean that they put autografts in conduits for the control? There is no explanation of what this means.
10. The discussion goes into a superficial listing of techniques, just stating that some were effective. There is not an in-depth discussion. Were the tests done correctly? Which tests were “better”? This depends on many factors and would be defined by things like what version of the test was done, what statistics were used and were the numbers of animals adequate?. Also, there are complexities to each technique. For example, the SFI has a very limited use now in PN repair because if the repair is successful, many animals develop contractures (due to uneven motor recovery) and then that and the SSI are useless. So that is why the BBB has been tried. But there are issues with that assay also. There are many excellent reviews in the literature that do a very thorough job of analyzing how good each test is in studying nerve repair. That sort of article could be quoted or discussed here.
11. Section 4.3 discusses trauma mechanisms. But the authors stated in the very beginning that they only looked at neurotmesis. So why did they include articles on nerve crush injuries (axonotmesis)?
12. Don’t use the word clinical to describe the studies that you are reporting on. This was used in the abstract: “Only experimental clinical trials ….were included.” Clinical refers to human studies. All murine studies are pre-clinical. Very confusing. Omit the word clinical to refer to these studies you are describing.
13. Define abbreviations, like: CNAP, RNL, Lv-ADSCS, 34a-ADSCS (the last two were in lines 142-143).
14. Don’t include small details of one experiment without describing it in appropriate detail. Why tell us about Lv- vs 43a- cells if you don’t describe the whole experiment? What is the point? If the method was adequate then you should have seen that it was successful at statistically differentiating between groups that differed by X% between groups, using x number of animals at a power of >0.8, then presumably the method is adequate.
Comments on the Quality of English Language
The English is fine, only minor edits were seen.
Author Response
Dear Reviewer,
Thank you for your thorough review and valuable feedback on our manuscript titled "Adipose Tissue Derivatives in Peripheral Nerve Regeneration After Transection: A Systematic Review." We have carefully considered each of your suggestions and have made the necessary revisions to address the issues raised. Below, we provide detailed responses to your comments and outline the specific changes made to the manuscript.
- The title and the abstract are very misleading and do not represent what was done. Both suggest that the authors are looking at ALL types of repair strategies for peripheral nerve repair. And then their wording suggests that they found that studies using adipose-derived stem cells (I will abbreviate ASCs) were “most common”. This implies that they looked for papers that used all types of nerve repair strategies and then within this, they found that the most used ASCs. Instead, they purposefully set out to search ONLY for ASCs in their search terms. This needs to be made clear in both the title and the abstract.
- Changes have been made to the title and abstract to better clarify the objectives of the proposed study. The search criteria for this review aimed to find articles that used adipose tissue derivatives to assess a better or worse rate of nerve regeneration in murine peripheral nerves subjected to nerve transection. Thus, the use of ASCs was the most commonly found among the various types of adipose tissue derivative use.
- What was the purpose of this review? The authors do not describe the criteria that they used to go from 273 articles down to 19. This has to be given, to understand why this review was written.
- The objectives of the present study have been better clarified in the title, abstract, and at the end of the introduction. The MeSH terms used for the search strategy as well as the inclusion and exclusion criteria of the reviewed articles are better explained in sections 2.1 and 2.2 of the Methodology. PRISMA flow diagram was updated to include exclusion reasons.
- Did the authors only look at ASCs? In Fig. 8 they list many different types of interventions. This includes one article that used BMSCs. There is a very extensive literature on BMSCs and their use in nerve repair. The same for MSCs in general and ASCs are one type of MSCs. If these fields were indeed included in their search, there is no way that they could come up with just one article. The search strategy did indeed include MSC as a term. But that is not one inter-vention listed in the graphs. ASCs are one type of MSC, and are often referred to as MSCs.
- The "Figure 8" and “Figure 9” have been removed to avoid confusion. Among the article selection criteria, it is stated that only articles utilizing adipose tissue derivatives in at least one evaluated subgroup were included. Thus, there are studies that assessed the potential of ASCs as a raw material for adipose tissue derivatives. A few articles evaluated the use of other types of MSCs in other comparator subgroups. These changes are aimed at facilitating understanding.
- Over what time span did the authors search? The authors say indefinite time frame, but they should give a more definitive time frame. Presumably they started in 2005. When did the search go until?
- A change has been made to the methodology to clarify the evaluated period.
- This is definitely NOT a thorough identification of all the articles that have used ASCs for peripheral nerve re-pair. Just a few examples are:
- A recent review of adipose tissue stem cells in peripheral nerve regeneration (Rhodes et al., J Neurosci Res. 2021;99:545–560. https://doi.org/10.1002/jnr.24738) lists Rhodes, cites 17 papers between 2016 and 2021. So if they could find 17 over 5 years, how did the authors find only 19 over a longer time period? Presumably the authors started their search in 2005, which is when they say that the first lab did this (Kokai et al. in 2005 from the K. Marra lab). (this is not my lab).
- The authors cite papers from well-known and prestigious labs: from K. Marra’s lab (Ref 37, Kokai et al.) and from the lab that included Di Summa, Kalbermatten and Terenghi, together at one point in Switzerland. Each of these three authors then separated, and some moved to other countries. However, only one paper is cited from, for example, Di Summa. All of these authors have multiple publications that describe ASCs in nerve repair. Example: an excellent review of the potential of ASCs for use in nerve regeneration was published in 2013 with Terenghi as one author. (These are not my labs).
- The present study reviews articles that used adipose tissue derivatives to assess a better or worse rate of nerve regeneration in murine peripheral nerves subjected to nerve transection. Thus, the use of ASCs was the most commonly found among the various types of adipose tissue derivatives use. Changes have been made to the title, abstract, introduction, and methodology to facilitate understanding of the objective of this review. Our search and article selection criteria differ from the reviews cited above, mainly regarding the evaluation only of nerve transection. The same applies to the need for evaluating nerve regeneration by different functional, histological, or imaging methods in the reviewed studies. Additionally, a better explanation of what we consider as adipose tissue derivatives has been added to the methodology to avoid confusion with the use of ASCs.
- The authors state in the Discussion, page 7, line 94, that a search on PubMed for ASCs shows an increasing trend of articles. Why didn’t the authors use PubMed as one of their sources of articles?
- The search was conducted on PubMed. A change was made to the “methodology” section to resolve this confusion.
- Section 4.2 discusses Fat Grafting and Stem Cells. There is an extensive literature that explores the use of grafting pieces of fat over and around nerve conduits for improving nerve repair. Is this what the authors are talking about?
- The title of section 4.2 was inadequate and was updated to “4.2. Adipose tissue derivatives”
- Figures 8 and 9 are almost identical. But I do not understand what the difference is between them. Especially Fig. 9. It supposedly lists: Materials used in nerve repair by “total number of nerves severed.” What does that mean?
- Figure 8 and 9 were removed. A new “Table 1” Was elaborated detailing the interventions of each study. We believe that this is a more interesting approach.
- The graphs are confusing in other ways. There are two colors for articles that presumably used conduits and others that did not. In Fig. 7, one paper used autografts as the control condition, but the autograft bar is labeled orange, meaning, I suppose, that it used conduits? Do you mean that they put autografts in conduits for the control? There is no explanation of what this means.
- Figure 7. was removed. Regarding the autograft, the article that utilized an autograft as a control was Kappos et al. (2015) (DOI: 10.1089/scd.2014.0424). This is an excerpt from their methods section:
“The sciatic nerve injury model was used, creating a 10- mm gap in the left nerve of female Sprague-Dawley rats (7 groups of 7 animals, 8 weeks old) that was bridged through a biodegradable fibrin conduit by the microsurgical suture technique. 1 Mio of each of the following cells was introduced into the conduits: rASCs, differentiated rASCs (drASCs), rat Schwann cells, human (h-)ASCs from the superficial and deep abdominal layer, as well as human SVF. As a control, we resutured a 10-mm cut nerve segment backward as an autograft.”
Our understanding from this explanation is that the nerve graft was placed inside a fibrin conduit, hence why the orange label.
- The discussion goes into a superficial listing of techniques, just stating that some were effective. There is not an in-depth discussion. Were the tests done correctly? Which tests were “better”? This depends on many factors and would be defined by things like what version of the test was done, what statistics were used and were the numbers of animals adequate?. Also, there are complexities to each technique. For example, the SFI has a very limited use now in PN repair because if the repair is successful, many animals develop contractures (due to uneven motor recovery) and then that and the SSI are useless. So that is why the BBB has been tried. But there are issues with that assay also. There are many excellent reviews in the literature that do a very thorough job of analyzing how good each test is in studying nerve repair. That sort of article could be quoted or discussed here.
- The objective of the article is to provide an overview of the current experimental assays present in the literature, as well as their "state of the art". The present study aims to serve as a guide for future production of experimental assays and as a tool for researchers to better assess the types of adipose tissue derivatives used as well as which models of nerve regeneration assessment were employed. We hope that the addition of the new "Table 1" clarifies all of this.
- Section 4.3 discusses trauma mechanisms. But the authors stated in the very beginning that they only looked at neurotmesis. So why did they include articles on nerve crush injuries (axonotmesis)?
- Articles studying crush injuries were not included in our study. Section 4.3 was optimized for clarity.
- Don’t use the word clinical to describe the studies that you are reporting on. This was used in the abstract: “Only experimental clinical trials ….were included.” Clinical refers to human studies. All murine studies are pre-clinical. Very confusing. Omit the word clinical to refer to these studies you are describing.
- Modification made throughout the manuscript to "experimental assays".
- Define abbreviations, like: CNAP, RNL, Lv-ADSCS, 34a-ADSCS (the last two were in lines 142-143).
- The addition of the new “Table 1” promises to better describe all the terms and abbreviations used in the studies.
- Don’t include small details of one experiment without describing it in appropriate detail. Why tell us about Lv- vs 43a- cells if you don’t describe the whole experiment? What is the point? If the method was adequate then you should have seen that it was successful at statistically differentiating between groups that differed by X% between groups, using x number of animals at a power of >0.8, then presumably the method is adequate.
- Modification made in the discussion to avoid confusion and not change the focus of the statement made.

Reviewer 3 Report
Comments and Suggestions for Authors
The authors report a systematic review of the literature for peripheral nerve injury in murine models and the multiple adipose derivatives used for nerve regeneration.
Some minor revisions require attention:
Introduction last paragraph – please be more specific in the objective of this systematic review. Because you excluded all papers that did not use fat derivatives after neurotmesis, I think you should rephrase your primary aim with regards to this aspect (example: to evaluate the role of fat derivatives after neurotmesis in murine models subjected to PNI…)
Page 3 Line 9-14 – I wonder whether some articles might have been missed with the limited keywords you selected. There are multiple ways to describe “peripheral nerve injury” that were not included in your mesh terms
Page 3 Line 29-30 – this belongs in the results section (“…19 studies were included…”)
Remove Figure 2 or Figure 3 (duplicate of the same information)
Page 6 line 55 – include the standard deviations or max-min for each average that you report
Figure 4, 5, 6, 7, 8, and 9 – in a systematic review, the reader is less interested by the number of articles than by the total number of participants. Please create new graphs where the Y axis is the total number of subjects (rather than the total number of articles)
Author Response
Dear Reviewer,
Thank you for your thorough review and valuable feedback on our manuscript titled "Adipose Tissue Derivatives in Peripheral Nerve Regeneration After Transection: A Systematic Review." We have carefully considered each of your suggestions and have made the necessary revisions to address the issues raised. Below, we provide detailed responses to your comments and outline the specific changes made to the manuscript.
Abstract: 10.16 change with 10.2
Modification made as requested.
- Introduction last paragraph – please be more specific in the objective of this systematic review. Because you excluded all papers that did not use fat derivatives after neurotmesis, I think you should rephrase your primary aim with regards to this aspect (example: to evaluate the role of fat derivatives after neurotmesis in murine models subjected to PNI…)
- Modification made in the introduction to better specify the objectives of the proposed study. Changes were made to the title and abstract to reduce possible confusion.
- Page 3 Line 9-14 – I wonder whether some articles might have been missed with the limited keywords you selected. There are multiple ways to describe “peripheral nerve injury” that were not included in your mesh terms
- We tested different descriptors to assemble the search strategy and concluded that using “Peripheral nerve injuries” [All Fields] OR “denervation” [All Fields] was the optimal quiry that most balanced sensitivity and specificity of the results. Other attemps retrieved a few more studies, but at great cost of specificity.
- Page 3 Line 29-30 – this belongs in the results section (“…19 studies were included…”)
- Modification made as requested.
- Remove Figure 2 or Figure 3 (duplicate of the same information)
- “Figure 2” was removed. Figure 3 was optimized for readability.
- Page 6 line 55 – include the standard deviations or max-min for each average that you report
- Standard deviations were included in each average.
- Figure 4, 5, 6, 7, 8, and 9 – in a systematic review, the reader is less interested by the number of articles than by the total number of participants. Please create new graphs where the Y axis is the total number of subjects (rather than the total number of articles)
- We have reconsidered the purpose of each Figure and have eliminated Figures 2, 4, 5, 7, 8, and 9. We hope that the addition of the new “Table 1” will be a more interesting approach.

Round 2
Reviewer 2 Report
Comments and Suggestions for Authors The authors have adequately addressed the comments of the review. Note a few writing mistakes that the editors can catch. One of importance is a misspelling of the word histological in the graphical abstract. ( hystological).Comments on the Quality of English Language The authors have adequately addressed the comments of the review. Note a few writing mistakes that the editors can catch. One of importance is a misspelling of the word histological in the graphical abstract. ( hystological).